# Distinct Effects of Stereotactically Injected Human Cerebrospinal Fluid Containing Glutamic Acid Decarboxylase Antibodies into the Hippocampus of Rats on the Development of Spontaneous Epileptic Activity

**DOI:** 10.3390/brainsci10020123

**Published:** 2020-02-22

**Authors:** Bernd Frerker, Marco Rohde, Steffen Müller, Christian G. Bien, Rüdiger Köhling, Timo Kirschstein

**Affiliations:** 1Oscar Langendorff Institute of Physiology, University of Rostock, 18057 Rostock, Germany; 2Department of Irradiation Therapy, University Hospital of Rostock, 18059 Rostock, Germany; 3Epilepsy Center Bethel, Krankenhaus Mara, 33617 Bielefeld, Germany; 4Center of Transdisciplinary Neurosciences Rostock, University of Rostock, 18147 Rostock, Germany

**Keywords:** glutamic acid decarboxylase, autoimmune limbic encephalitis, NMDA and GABA receptor, recurrent epileptiform discharges, human cerebrospinal fluid, hippocampus of rats, field potential recording

## Abstract

*Background*: The conversion of glutamic acid into γ-aminobutyric acid (GABA) is catalyzed by the glutamic acid decarboxylase (GAD). Antibodies against this enzyme have been described in neurological disorders, but the pathophysiological role of these antibodies is still poorly understood. We hypothesized that anti-GAD autoantibodies could diminish the GABA content in the slice and facilitate epileptic activity. *Methods*: Cerebrospinal fluids (CSF) from two patients containing anti-GAD (A and B) were injected into the rat hippocampus in vivo. Hippocampal slices were prepared for electrophysiological field potential recordings in order to record recurrent epileptic discharges (REDs) in the CA1 region induced by the removal of Mg^2+^ and/or by adding gabazine. As control groups, we injected an anti-GAD-negative human CSF or saline solution, and we used non-operated naive animals. *Results*: RED frequencies were significantly higher in the Mg^2+^-free solution than in the gabazine-containing solution. The average frequency of REDs in the last 10 min and the average duration of REDs in the last 5 min did not show significant differences between the anti-GAD-B-treated and the control slices, but in the Mg^2+^-free solution, anti-GAD-A had significantly higher epileptic activity than anti-GAD-B. *Conclusions*: These results indicate that anti-GAD has distinct effects on the development of spontaneous epileptic activity.

## 1. Introduction

*Limbic encephalitis (LE)* is a syndrome composed of temporal lobe epilepsy, loss of short-term memory and psychiatric symptoms such as depression or irritability. Magnetic resonance imaging (MRI) shows unilateral or bilateral inflammation of the temporal lobe [1]. Initially, this disease was described by Brierley in 1960 [2] in patients with a subacute encephalitis and usually considered to be of paraneoplastic origin [3]. Later on, cases were reported to be associated with specific autoantibodies (*autoimmune limbic encephalitis, ALE)*, either to cell membrane antigens like the N-methyl-D-aspartate receptor (NMDA-R) or the γ-aminobutyric acid receptor (GABA-R) [4], as well as to intracellular antigens like Hu (anti-neuronal nuclear antibody-1) in patients with lung cancer [5], Ma2 (paraneoplastic antigen Ma2) in patients with testicular tumors [6] and CRMP5/CV2 (collapsin response mediator protein 5) in patients with thymomas [7]. In addition, antibodies to the intracellular enzyme glutamic acid decarboxylase (anti-GAD) have also been reported both in paraneoplastic [8] and non-paraneoplastic cases of LE [9].

*Antibodies against the glutamic acid decarboxylase (anti-GAD)* were first described in cases of stiff-person syndrome, often associated with type 1 diabetes mellitus [10,11]. Other neurological disorders associated with anti-GAD were the cerebellar ataxia [12] or some cases with intractable epilepsy [13,14]. A few years ago, anti-GAD was found in patients with LE [8,9].

*The glutamic acid decarboxylase (GAD)* is intracellularly localized in the GABAergic neurons of the central nervous system (CNS) or extracerebrally in the β-cells of the pancreas. It catalyzes the conversion from glutamate into GABA in a pyridoxal phosphate-dependent decarboxylation. Therefore, it is mainly responsible for the production of GABA, the most important inhibitory neurotransmitter in the CNS [15]. Although this enzyme is believed to be the primary target of anti-GAD, the pathophysiological role of these antibodies is poorly understood. Previous studies suggested an inhibition of GABA-synthesis [16] or reduced synaptic GABAergic transmission [17] as a result of anti-GAD. As a consequence, one would argue that there may be a lower concentration of GABA that would facilitate the development of seizures due to increased neuronal excitability. For example, using a whole-cell patch-clamp technique, Ishida et al. [18] confirmed that continuous perfusion of rat cerebellar slices with anti-GAD caused a selective suppression of GABAergic transmission. However, those studies have been performed in vitro, and the role in vivo is less clear. Recently, Hackert et al. [19] performed a hippocampal stereotactic injection of human cerebrospinal fluid (CSF) containing anti-GAD in vivo. They found that the applied antibodies did not alter GABAergic synaptic transmission in intracellular and patch-clamp recordings in vitro. To test for the propensity to develop hyperexcitability directly, we performed field potential recordings in order to record recurrent epileptic discharges (REDs) in the CA1 region of hippocampal slices induced by the removal of Mg^2+^ and/or adding gabazine (5 µM) after a stereotactic injection of anti-GAD. 

## 2. Materials and Methods

### 2.1. Stereotactic Intrahippocampal Injection in Vivo

Before mounting on a stereotactic frame (Narishige, Tokyo, Japan), 2-month-old female Wistar rats (Charles River Laboratories GmbH, Sulzfeld, Germany) were anesthetized with S-ketamine (100 mg/kg IP (intraperitoneal)) and xylazine (15 mg/kg IP). The injection sites were calculated relatively to the bregma (+ 5.2 mm posterior, ± 4.3 mm lateral, + 4.8 mm deep), as previously reported [19,20,21]. After the trepanation, bilateral injection of 5 µL human non-diluted CSF (into each hippocampus) was performed using a Hamilton syringe (75 N, Hamilton AG, Bonaduz, Switzerland) in 10 steps of 0.5 µL every two min. After the last aliquot of 0.5 µL, the syringe was left in situ for another two minutes. CSF samples containing anti-GAD were obtained from two patients (both female, 18 years old and 27 years old) with temporal lobe epilepsy due to limbic encephalitis. The GAD-antibodies were detected by indirect immune-techniques by congruent GAD65 specific signals on mouse brains, transfected cells and immune-dot-blot. Titration was done on cell-based assays in dilution steps of 1:2 and multiples (see [19]). The procedure was described by Christian G. Bien [21,22]. The titers were 1:1000 (anti-GAD A) and 1:250 (anti-GAD B). As control groups, we injected either a CSF sample from a patient with epilepsy (absence of anti-GAD) or a saline solution (NaCl 0.9%). In order to control for the surgery, field potential recordings were also performed in non-operated, naive animals as a third control group (Table 1). Since we only had a small sample of anti-GAD-A, experiments II and III were carried out only with anti-GAD-B. Following postoperative pain management (metamizole 100–150 mg/kg), the rats were allowed to recover in an atmosphere with enhanced oxygen fraction (4–5 L/min in an 8 L glass vessel). 

To confirm the diffusion of the injected agents (CSF samples containing anti-GAD A and B, anti-GAD-negative human CSF and saline solution) as previously reported [20], we performed experiments using an injection of an immunofluorescent marker dye. After injection, cryostat sections of the brain were covered with ProLong® Gold antifade reagent with DAPI (Invitrogen/ThermoFisher Scientific, Carlsbad, CA, USA). Evaluation was done using the Leica DMI6000B microscope and LAS AF software (Figure 1).

### 2.2. Hippocampal Slice Preparation

One day after the stereotactic injection, the animals were decapitated in order to prepare the hippocampal transversal brain slices. Deep anesthesia was realized with diethyl ether (Baker, Deventer, Netherlands) and controlled by the absence of pain reflexes. The rats were then decapitated and the brains were quickly dissected out and submerged into oxygenated iced sucrose-based dissection fluid (in mM: NaCl 87, sucrose 75, NaHCO_3_ 25, KCl 2.5, NaH_2_PO_4_ 1.25, CaCl_2_ 0.5, MgCl_2_ 7, glucose 10; pH = 7.4, osmolality 300–320 mosmol/kg H_2_O). Hippocampal slices (400 µm) were then prepared by using a vibratome (Integraslice 7550 MM, Campden Instruments, UK) and transferred into a self-made storage chamber containing standard **a**rtificial **c**erebro**s**pinal fluid (ACSF, in mM: NaCl 125, NaHCO_3_ 21, KCl 3, NaH_2_PO_4_ 1.25, CaCl_2_ 2.5, MgCl_2_ 1, glucose 13; pH = 7.4, osmolality 299-306 mosmol/kg H_2_O). The fluids were continuously gassed with 95% O_2_ and 5% CO_2_ to maintain the pH at 7.4. The slices adapted for at least one hour in the storage chamber before being used for field potential recordings.

### 2.3. Field Potential Recordings

The slices were placed into an interface chamber (BSC-HAT, Harvard Apparatus, USA) and continuously perfused with standard ACSF. The recording temperature was maintained at 32 °C (TC-10, npi electronics, Germany). Electrodes (PIP5 puller, HEKA Elektronik, Germany; filled with standard ACSF) were placed directly above the CA1 region without perforating the hippocampus. Using the standard ACSF, the hippocampal slices adapted for 20 min. Then, the recordings were started using the standard ACSF again to obtain a baseline confirming the absence of spontaneous epileptic activity at the beginning. After 20 min, the standard ACSF was switched to a modified one to induce REDs, either by Mg^2+^-free ASCF [23] or by adding gabazine 5 µM to standard ACSF [24]. In some experiments, we combined both methods, i.e., we added gabazine 5 µM to the Mg^2+^-free ACSF (Table 1). The recordings were continued for a further 130 min. In order to be recognized as a RED, the potential deflections were required to have an amplitude of at least 0.1 mV. The duration was defined as $\Delta t=\;\left|t_{1}-t_{2}\right|$ with t_1_ as the time point of the initial positive or negative deflection and t_2_ as the time point of completed repolarization (indicated by gray boxes in Figure 2). Since a RED may have more than one phase (i.e., multiple positive and negative deflections), completed repolarization was defined as the final decline back to baseline noise. 

### 2.4. Data Processing and Statistical Analysis

The data were processed using an analog-to-digital converter (Power 1401, CED, UK) and Signal 2.16 Software (CED, UK). A qualitative data analysis was performed in order to identify artifacts and to determine the duration and morphology of REDs. A quantitative data analysis was carried out by manually counting the registered REDs. Statistical analyses were performed using non-parametric tests (a Mann–Whitney U-test or a Kruskal–Wallis ANOVA rank test). The level of significance was set to 0.05. The data are presented as means ± standard error of the mean (SEM). 

### 2.5. Ethics Approval Statement

All animal procedures were approved by the local Animal Welfare committee and conformed to national and international guidelines on the ethical use of animals (European Council Directive 86/609/EEC, approved by the Regional Authorities of Agriculture, Food Safety and Fishery Mecklenburg-West Pomerania, no. 7221.3-1.1-017/11). All efforts were made to minimize animal suffering and to reduce the number of animals used. The patients gave their written informed consent to use the CSF samples for scientific purposes in accordance with the declaration of Helsinki. 

## 3. Results

In the present study, we aimed to address the question of whether autoantibodies against GAD would facilitate hyperexcitability in an acute epilepsy model. We hypothesized that anti-GAD autoantibodies could diminish the GABA content in the slice and thereby facilitate epileptic activity. To this end, we induced recurrent epileptiform discharges by three different paradigms (Table 1). 

### 3.1. Marker Dispersion in the Hippocampus

Firstly, we have confirmed the injection site (see Figure 1, red plus) using an immunofluorescent marker dye as previously described [20]. After injection, the marker disperses along the vessels into the CA1-region of the hippocampus. Thus, it is obvious that the CSF containing anti-GAD can reach the CA1-region.

### 3.2. Morphology of REDs

We analyzed the *morphology of REDs* in all experimental groups. Depending on the pattern and duration of the potentials, we could distinguish four different types of REDs: interictal-like single monomorphic (Figure 2A) discharges, repetitive monomorphic (Figure 2B) and polymorphic discharges (Figure 2C) as well as ictal-like discharges (Figure 2D). The interictal-like discharges (Figure 2A-C) were routinely observed in all recordings irrespectively from the RED-inducing ACSF (Table 1). In contrast, ictal-like discharges were rare and only obtained in the gabazine-containing ACSF (5/47), but never in the Mg^2+^-free ACSF (0/33, *p* = 0.0663; χ^2^ test). In summary, we did not find significant differences in the morphology of epileptic discharges among the different animal groups (i.e., the anti-GAD-treated [A and B] or control groups) or different RED-inducing solutions.

### 3.3. Average Frequency of REDs

Then, we counted the *number of REDs* in all recordings. At the beginning of each experiment using the standard ASCF, the absence of spontaneous activity was confirmed in all experiments (Figure 3A,B). Upon changing the standard ACSF to the modified ACSF (Table 1), REDs occurred after 20–30 min and reached a plateau after 90–100 min (Figure 3A,B). Hence, we calculated the *average RED frequency* for the last 10 min to compare slices from the anti-GAD-treated rats and the controls (Figure 3C). While the slices bathed in the Mg^2+^-free ACSF generally showed higher RED frequencies than the gabazine-treated slices (Mg^2+^-free versus gabazine: *p* = 0.008; Mg^2+^-free/gabazine versus gabazine: *p* = 0.017; 2-way-ANOVA and Tukey post-hoc test), no significant differences were obtained between the anti-GAD-B and their corresponding control slices (Mann–Whitney U-test or Kruskal–Wallis ANOVA on ranks, Figure 3C). Interestingly, in the Mg^2+^-free solution, the anti-GAD-A showed significantly higher epileptic activity compared to the corresponding anti-GAD-B (anti-GAD-A: 25 ± 6 per min, anti-GAD-B: 14 ± 3 per min, *p* = 0.04, Mann–Whitney Rank Sum Test). Naive rats showed a similar incidence of epileptic activity within each experimental group, thus an influence due to the surgical procedure is very unlikely. 

### 3.4. Average Duration of REDs

We also determined the *average duration of all REDs* during the last 5 min of each recording (Figure 3D). However, there were again no significant differences between the anti-GAD-treated groups and the control groups (*p* > 0.3 for all comparisons, Mann–Whitney U-test or Kruskal–Wallis ANOVA on ranks). The average duration of anti-GAD-A treated slices was equal to the anti-GAD-B in the Mg^2+^-free solution (*p* = 0.483). Since the duration of individual REDs showed substantial variance (Figure 2), we calculated the *cumulative time* of the slices showing epileptic discharges during the last 5 min of each experiment. There were again no significant differences between the anti-GAD-B-treated and control groups (Mg^2+^-free: anti-GAD-B: 73.5 s ± 8.9 s, control-groups: 54.9 s ± 20.0 s, *p* = 0.547; gabazine: anti-GAD-B: 35.4 s ± 7.9 s, control-groups: 21.7 s ± 3.9 s, *p* = 0.319; Mg^2+^-free and gabazine: anti-GAD-B: 79.6 s ± 8.7 s, control-groups: 76.2 s ± 12.7 s, *p* = 0.443; one-way-ANOVA). In the Mg^2+^-free solution, the anti-GAD-A showed a significantly higher incidence of REDs than the anti-GAD-B. Thus, we compared the cumulative duration, and indeed, the cumulative duration of anti-GAD-A was significantly longer (123.2 s ± 14.8 s vs. 73.5 s ± 8.9 s, *p* = 0.005).

## 4. Discussion

The presence of autoantibodies against glutamic acid decarboxylase in patients with epilepsy and encephalitis is probably pathophysiologically relevant. Certainly, the most parsimonious assumption is a reduced GABA content due to an inhibition of the GABA-synthetizing enzyme GAD [16]. In addition, other pro-epileptic effects of anti-GAD are reported in the literature: a reduced inhibition of presynaptic glutamate release [25], a cross-reaction with GABA binding sites on the neuronal surface [16] and, finally, a reduced NO-dependent suppression of glutamate synthesis [26].

In order to study potentially pro-epileptic effects of a stereotactic anti-GAD injection on hyperexcitability, we first used a Mg^2+^-free ACSF solution to induce acute epileptic activity by the NMDA receptor disinhibition in the slices [23]. Naive rats showed the same results as control-injected rats, indicating that confounding effects from anesthesia and/or surgery are unlikely. No significant differences between anti-GAD-B-treated and the control-injected slices were obtained, suggesting that NMDAR-dependent epileptic activity was not altered by anti-GAD-B. These results are compatible with a normal GABA content in the slices, but they also suggest that epileptic activity due to a disinhibition of the NMDA receptors could overcompensate a potentially reduced GABA level. Therefore, we used the GABA_A_R blocker gabazine alone to induce epileptic activity, and indeed we found significantly lower frequencies of epileptiform discharges compared to the Mg^2+^-free condition. There were again no significant differences between the anti-GAD and control slices. The third paradigm adding gabazine to the Mg^2+^-free ACSF solution revealed similar epileptic activity as in the experiments using the Mg^2+^-free ACSF solution alone, confirming that NMDAR disinhibition by the Mg^2+^-free condition had a higher potency in producing hyperexcitability than the GABA_A_ receptor blockade. 

Interestingly, ictal-like epileptic activity did not appear in the gabazine-free solution-treated hippocampal slices, neither with anti-GAD-A nor with anti-GAD-B. In particular, if anti-GAD led to a reduced GABA content in the slices, one would expect ictal-like activity to occur in the slices from anti-GAD-treated animals bathed in the Mg^2+^-free solution. 

However, in the Mg^2+^-free ACSF, anti-GAD-A induced significantly more REDs than the corresponding anti-GAD-B. These results are compatible with a suppression of GABA in hippocampal neurons, similar to the results of other study groups. Using whole-cell voltage clamp recordings, Ishida et al. [18] showed that continuously perfused rat cerebellar slices with ACSF containing anti-GAD led to a presynaptic inhibition of cerebellar GABAergic transmission. In addition, Vianello et al. [27] revealed an increased spontaneous activity of hippocampal neurons in culture caused by the suppression of inhibitory potentials mediated by anti-GAD. Nevertheless, the anti-GAD-B was not different from the corresponding control groups. These negative findings are consistent with previous reports showing intact GABAergic transmission in anti-GAD-treated tissue [19,28].

Specific immunoreactivity of anti-GAD with the intracellular enzyme has been documented in a number of studies concerning the intrathecal injection of antibodies [29], incubation of hippocampal neurons with anti-GAD-positive sera [27,30], and staining cerebellar neurons with anti-GAD-containing CSF [17]. Furthermore, anti-GAD might recognize different epitopes on the GAD enzyme, and this might help explain why anti-GAD is associated with different diseases like the stiff-person-syndrome, cerebellar ataxia, intractable epilepsy, limbic encephalitis and type 1 diabetes mellitus [26,30]. 

In summary, we found distinct effects of stereotactically injected human CSF containing anti-GAD into the hippocampus of rats on the development of spontaneous epileptic activity. There is accumulating evidence that anti-GAD has no effect in vivo such as the flow cytometric detection of cytotoxic CD8-cells [31]. It is an intriguing idea that further factors in the patient sera might be present in addition to anti-GAD [32,33]. The pathophysiological role of anti-GAD remains controversial.

Some limitations need to be discussed. Firstly, dilution of injected antibodies cannot be excluded, albeit previous studies with CSF from anti-NMDAR encephalitis patients provided evidence that sufficient antibody concentrations were achieved to induce biological effects [21,34]. Secondly, we cannot exclude that the limited sample size, and hence limited power, may have masked a small difference between the anti-GAD-treated and control tissue. 

## Figures and Tables

**Figure 1 brainsci-10-00123-f001:**
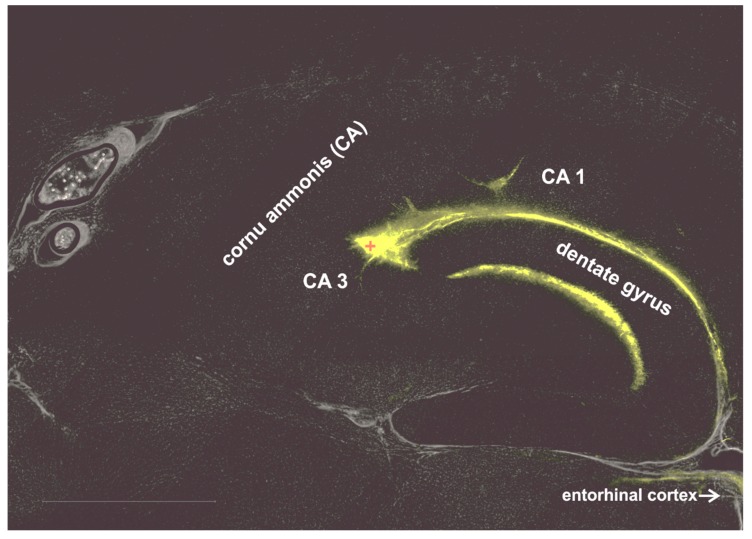
Marker dispersion in the hippocampus. The immunofluorescence image shows that the injected marker disperses from the injection site (CA3-region) into the CA1-region. (injection site denoted by the red +). CA = Cornu Ammonis.

**Figure 2 brainsci-10-00123-f002:**
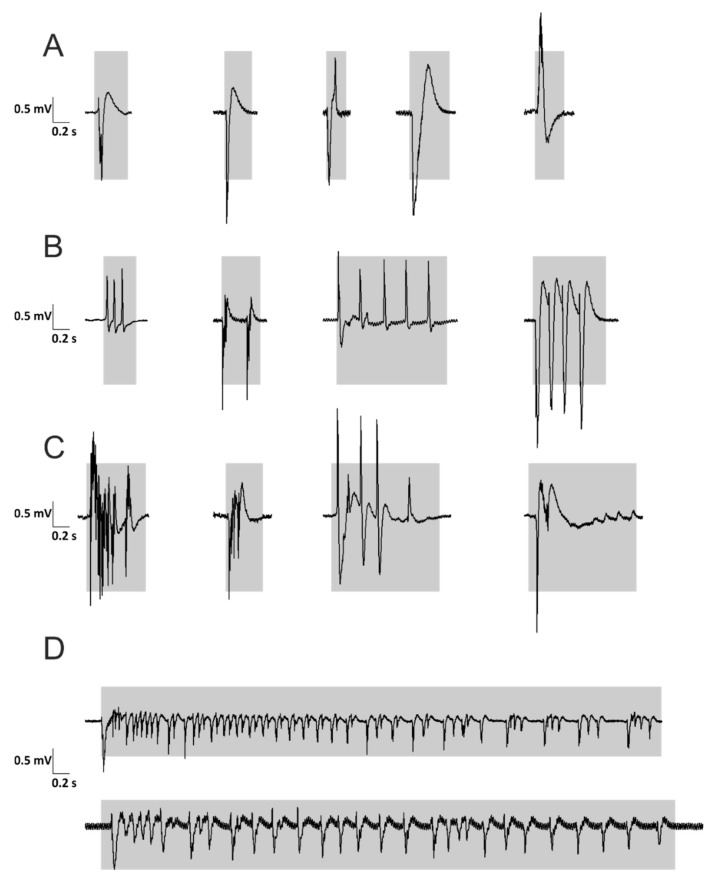
Morphology of REDs in all experimental groups. Depending on the pattern and duration of the potentials, we could distinguish four different types of REDs: interictal-like single monomorphic (**A**), repetitive monomorphic (**B**), and polymorphic discharges (**C**) as well as ictal-like discharges (**D**). Ictal-like discharges were only obtained in gabazine-containing ACSF. The durations of REDs are highlighted in gray.

**Figure 3 brainsci-10-00123-f003:**
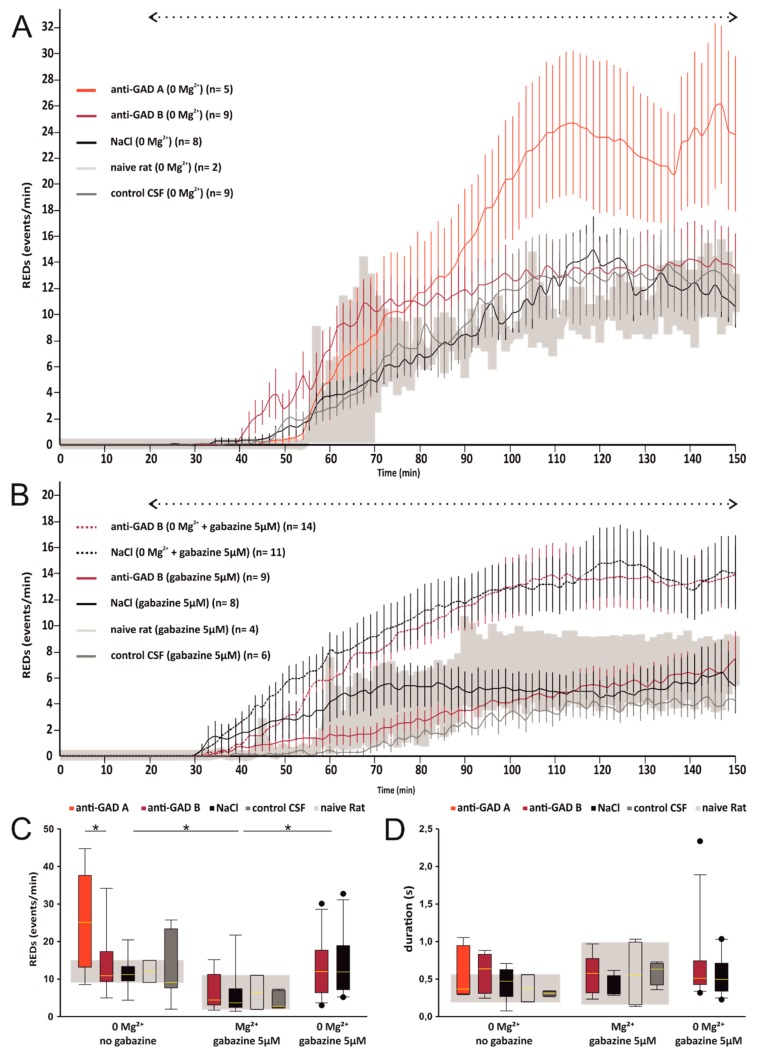
Average RED frequency and average duration. All slices showed no spontaneous activity at the beginning of each recording. Upon changing standard ACSF to the modified one, REDs occurred after 20–30 min (**A,B**). Average RED frequency for the last 10 min revealed no significant differences between the anti-GAD-B-treated and corresponding control tissues, but slices in the Mg^2+^-free ACSF generally showed significant higher RED frequencies than the gabazine-treated slices (* in panel **C**). Furthermore, in the Mg^2+^-free ASCF the anti-GAD-A had significantly higher RED frequencies than anti-GAD-B. Panel **D** shows the average duration of all REDs during the last 5 min of each recording, and again there were no significant differences. The mean frequency and mean duration of naive slices are highlighted in (light) gray. An influence due to the surgical procedure is very unlikely. Outliers are plotted as black dots (panel **C** + **D**). In Mg^2+^ free and gabazine containing solution we found one slice treated with anti-GAD B that showed a higher average duration (t_average_ = 2.33 s) compared to the average duration of the corresponding slices (panel **D**). However, the cumulative time of this slice (t_cumulative_ = 76.6 s) was equal to the other slices treated with anti-GAD B (*p* = 0.4). The other outliers are close to the maximum and minimum and have therefore not been evaluated separately.

**Table 1 brainsci-10-00123-t001:** Experimental groups. Number of slices is given in parentheses.

Experiment	Anti-GAD Group	Control Groups	Recording Baseline (20 min)	Recording REDs (130 min)
I	anti-GAD-A (*n* = 5) anti-GAD-B (*n* = 9)	NaCl (*n* = 8) naive rat (*n* = 2) control CSF (*n* = 9)	ACSF standard	Mg^2+^ free
II	anti-GAD-B (*n* = 9)	NaCl (*n* = 8) naive rat (*n* = 4) control CSF (*n* = 6)	ACSF standard	gabazine 5 µM
III	anti-GAD-B (*n* = 14)	NaCl (*n* = 11)	ACSF standard	Mg^2+^ free gabazine 5 µM

REDs: recurrent epileptic discharges, ACSF: artificial cerebrospinal fluid.

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
