# Peer review of "Distinct Effects of Stereotactically Injected Human Cerebrospinal Fluid Containing Glutamic Acid Decarboxylase Antibodies into the Hippocampus of Rats on the Development of Spontaneous Epileptic Activity"

_brainsci, 2020, doi:10.3390/brainsci10020123_

Round 1

Reviewer 1 Report

The authors have thoroughly revised the manuscript and answered all the concerns raised in the first submission. 

Author Response

Dear Reviewer,

thank you very much for your nice comment, we were very pleased. 

We have made small changes (highlighted in violet) and we hope that our manuscript can be accepted for publication now.

Mr. Frerker

Reviewer 2 Report

This resubmitted manuscript includes details in methodology concerning in vivo experiment which were not shown in the first version and newly obtained data on the effects of anti-GAD-A on REDs. The authors found that anti-GAD-A induced a significantly higher epileptic activity than anti-GAD-B. The results are interesting and this data enhanced the value of this manuscript.

On the other hand, there seems to be a number of mistakes in words in text and figures, and the statement of some sentences is unclear. Furthermore, the format of citations in References and text does not follow the publisher's instructions. These shortcomings should be fixed and improved. 

The parts to be corrected are given below.

Abstract:

line 22,  GAD-negative human CSF -> anti-GAD-negative human CSF

line 24,  gabazine-treated slices -> gabazine-containing solution

line 26,  anti-GAD-B and the different control groups -> anti-GAD-B-treated and the control slices

Materials and Methods

line 82,  What does CGB mean?   

            [22, 21] -> [21,22]

line 90,  Specific names of the injected agents should be indicated.

Results:

lines 169, 226, 229,    0-Mg2+ -> Mg2+-free

line 190,  anti-GAD-B and -> anti-GAD-B-treated and

lines 204, 205,   0 Mg2+ -> Mg2+-free

Discussion:

lines 251-252, incubation of hippocampal neurons GAD-positive  ->  incubation of hippocampal neurons with GAD-positive

lines 245-259,  This paragraph starts with "In summary,". However, the contents are not like summary but rather it seems like a part of introduction. Major findings of this study and their significance in the research field of temporal lobe epilepsy should be included in "summary" or "conclusion".

Figure 3. A and B

Labels of ordinate: (1/min) -> Does this mean (events/min) ?

"1/min" does not seem to be appropriate.

controle CSF -> control CSF

References

The format of References should be revised according to Introduction for Authors.

Author Response

Dear Reviewer, 

thank you very much for evaluating our revised manuscript. We are very pleased that we could enhance the value of our manuscript. And thank you very much for your corrections. The changes we made are highlighted in violet

Materials and Methods

line 82,  What does CGB mean?   

Comment: We apologize for using initials. CGB is one of the authors, Christian G. Bien. He provided the human CSF samples containing anti-GAD.

Figure 3. A and B

Labels of ordinate: (1/min) -> Does this mean (events/min) ? "1/min" does not seem to be appropriate. controle CSF -> control CSF

Comment: You are absolutely right. The labels don't make sense. We have accepted your proposal because it seems appropriate to us.

Discussion:

lines 245-259,  This paragraph starts with "In summary,". However, the contents are not like summary but rather it seems like a part of introduction. Major findings of this study and their significance in the research field of temporal lobe epilepsy should be included in "summary" or "conclusion".

Comment: We agree that our discussion had to be revised. We now summarize our major finding (distinct effects of anti-GAD...). We hope that our changes are suitable.

References

The format of References should be revised according to Introduction for Authors.

Comment: Thank you for this remark. We revised our references according to the Introduction for Authors. We only list two authors, and we use the abbreviations of the journals. 

Thank you very much for giving us the opportunity to improve our revised manuscript. We hope that the changes are suitable and our manuscript can be accepted for publication.

Mr. Frerker

This manuscript is a resubmission of an earlier submission. The following is a list of the peer review reports and author responses from that submission.

Round 1

Reviewer 1 Report

The present study reports that the average frequency of REDs recorded from hippocampal slices was unaffected by an in vivo pretreatment of the hippocampus with human CSF which contained anti-GAD autoantibodies. The author concluded in Abstract that anti-GAD did not facilitate the development of spontaneous (epileptic) activity.

In addition, the researchers had previously performed similar experiments (Ref. 18) in which they injected human CSF containing GAD65-antibodies in hippocampus in vivo and measured IPSP and IPSC in vitro. They got similar results as such that the pretreatment of the hippocampus with CSF containing GAD-65 autoantibodies in vivo had no effects on IPSP/IPSC later on in vitro.

Notably, an another research group had indicated, by using cerebellar slices, that continuous perfusion of CSF that contained anti-GAD autoantibodies caused a selective suppression of GABAergic transmission in vitro (Ishida K et al., Ann. Neurol. 46, 263-267, 1999). This group had carried out delicate serological tests to determine the titer of the anti-GAD autoantibody before the electrophysiological experiments and described the values in the literature. They had also indicated the binding efficacy of the anti-GAD autoantibody by applying it to western blot analysis and immunohistochemistry.

I feel that the author should have confirmed the efficacy of the in vivo injection of anti-GAD autoantibody into the hippocampus by using western blot analysis and/or immunohistochemistry one day after the stereotactic injection (the timing of the hippocampal slice preparation). As long as I understand from M&M in this report, the authors had not confirmed the hippocampal areas where anti-GAD autoantibody-bound neurons exist. Unless perfusing the anti-GAD autoantibody with perfusate, it seems very difficult to achieve constant antibody-binding levels in the hippocampus tissue during electrophysiological recording in vitro. These issues should be clarified.

Minor points:

line 21, hippcampal -> hippocampus

line 28, spontaneous activity -> spontaneous epileptic activity

line 75, titer 1:250 -> How was this value obtained?

line 75, Have you confirmed the ability of this CSF sample from a 27-year old patient to recognize a specific epitope of GAD? Presence of anti-GAD autoantibody in a CSF does not necessarily mean that it has suppressive effects of GAD activity. 

Author Response

Dear Editor, and Dear Reviewers,

thank you very much for reading and evaluating our manuscript. Your comments were very useful to improve our manuscript. In the meantime, we had the opportunity to evaluate a GAD sample from another patient. Our new data suggests that this antibody could facilitate epileptic discharges in hippocampal slices, similar to the results of Ishida et al (1999). We included these novel data into our revised manuscript. Moreover, we repeated experiments showing the diffusion of an injected immunofluorescence marker dye.

We highlighted our changes in yellow. We hope this will help you to understand our changes.

Response to Reviewer One

Comments and Suggestions for Authors

The present study reports that the average frequency of REDs recorded from hippocampal slices was unaffected by an in vivo pretreatment of the hippocampus with human CSF which contained anti-GAD autoantibodies. The author concluded in Abstract that anti-GAD did not facilitate the development of spontaneous (epileptic) activity.

In addition, the researchers had previously performed similar experiments (Ref. 18) in which they injected human CSF containing GAD65-antibodies in hippocampus in vivo and measured IPSP and IPSC in vitro. They got similar results as such that the pretreatment of the hippocampus with CSF containing GAD-65 autoantibodies in vivo had no effects on IPSP/IPSC later on in vitro.

Notably, an another research group had indicated, by using cerebellar slices, that continuous perfusion of CSF that contained anti-GAD autoantibodies caused a selective suppression of GABAergic transmission in vitro (Ishida K et al., Ann. Neurol. 46, 263-267, 1999). This group had carried out delicate serological tests to determine the titer of the anti-GAD autoantibody before the electrophysiological experiments and described the values in the literature. They had also indicated the binding efficacy of the anti-GAD autoantibody by applying it to western blot analysis and immunohistochemistry.

Comment: As an opportunity, we had a small sample from our co-author (CGB), which could only be used for some rats in our first experiment (anti-GAD A, n = 5 slices, recording REDs with Mg2+-free ASCF, see table 1). Unlike our previous results, this antibody facilitated the development of spontaneous epileptic activity, similar to the results of Ishida et al. These novel findings are shown in Figure 3 C and D (formerly Figure 2; we added a new figure). The procedure of determining the GAD-titer is now described in our revised manuscript. The values are mentioned (the titers were 1:250 and 1:1000) The samples were obtained from lumbar puncture. We only had a small sample size, and we were not able to perform immunohistochemistry. As also stated below, we have performed a validation study to confirm proper dispersion of the CSF within the hippocampus.

I feel that the author should have confirmed the efficacy of the in vivo injection of anti-GAD autoantibody into the hippocampus by using western blot analysis and/or immunohistochemistry one day after the stereotactic injection (the timing of the hippocampal slice preparation). As long as I understand from M&M in this report, the authors had not confirmed the hippocampal areas where anti-GAD autoantibody-bound neurons exist. Unless perfusing the anti-GAD autoantibody with perfusate, it seems very difficult to achieve constant antibody-binding levels in the hippocampus tissue during electrophysiological recording in vitro. These issues should be clarified.

Comment: The injection site was identical to our previous reports (Hackert et al [Frontiers in cellular neuroscience 2016;10:130], Kersten et al [Frontiers in neurology 2019;10:586], where antibody diffusion and effects were present. The coordinates we used are still valid: We injected an immunofluorescence marker dye. In our new figure (figure 1, revised manuscript), we show the marker dispersion into the CA-1 region of the hippocampus. Thus, it is obvious that the CSF containing anti-GAD can reach the CA1-region. The CSF-sample containing anti-GAD from our new patient indeed has effects (new data, figure 3), and definitely indicates the presence in the hippocampus, independently from the existence of anti-GAD or other pathological agents.

Minor points:

line 21, hippcampal -> hippocampus

Comment: Our final grammar checks obviously failed. Trying not to exceed a total amount of 200 words in our abstract, this was deleted. Regarding our novel data, we made some changes.

line 28, spontaneous activity -> spontaneous epileptic activity

Comment: This was corrected.

line 75, titer 1:250 -> How was this value obtained?

Comment: We deeply apologize that we did not mention the procedure of titer determination. The procedure is now described in our section 2.1 (Stereotactic intrahippocampal injection in vivo). The changes are highlighted in yellow.

line 75, Have you confirmed the ability of this CSF sample from a 27-year old patient to recognize a specific epitope of GAD? Presence of anti-GAD autoantibody in a CSF does not necessarily mean that it has suppressive effects of GAD activity. 

Comment: The reviewer is absolutely right that the presence of an antibody does not necessarily predict biological effects. Biochemical assays to validate anti-enzymatic effects (such as Lineweaver-Burk characteristics) are not feasible with the small sample size obtainable from lumbar puncture. However, our new data supports the view that CSF containing anti-GAD may have a suppressive effect. Our anti-GAD A facilitates spontaneous epileptic activity, whereas our anti-GAD B does not. We discuss this issue at the end of our manuscript.

Response to Reviewer Two

Comments and Suggestions for Authors

The study titled "Stereotactically injected human cerebrospinal fluid containing glutamic acid decarboxylase antibodies into the hippocampus of rats does not facilitate the development of spontaneous epileptic activity" by Frerker et al., is an interesting report and important to be published. The authors have shown using hippocampal slices and electrophysiological field potentials that contrary to the intuitive notion, GAD antibody does not facilitate the development of spontaneous activity. 

The manuscript is well written. However, there are some minor concerns: 

Comment: First, thank you very much for your evaluation of our manuscript. In the meantime, we had the opportunity to improve our manuscript. We received a CSF-sample from another patient. As a result, this anti-GAD facilitated the spontaneous epileptic activity in the hippocampus. This is an important finding, since anti-GAD does not necessarily mean that it has suppressive effects on the GAD enzyme. For example, there might be different epitopes on the GAD enzyme, and this might help explain why anti-GAD is associated with different diseases. We hope that our new results improve our manuscript. The changes we made are highlighted in yellow.

1) Please write the full version of RED in the abstract instead of just the abbreviation. 

Comment: We apologize that we forgot to write the full version of REDs. Now, we have introduced this abbreviation.

2) It is not clear why the authors chose CA1 region for the recordings? Why not any other region of the Dentate? 

Comment: Stanton et al (1986) showed that perfusion of hippocampal slices with Mg2+-free solution induced spontaneous epileptic activity in the dentate gyrus. To our knowledge, in their experiments, the connections to the entorhinal were intact. Mody et al (1987) then showed that REDs are induced in the CA1- and CA3-region, whereas the dentate gyrus did not. In these cases, the connection to the entorhinal cortex was severed. This implicates that intact entorhinal cortex is necessary for the induction of spontaneous epileptic activity. From previous studies and the current study (new figure 1 in the revised manuscript) we know that the antibody dispersion includes the upper blade of the dentate, but rarely the lower blade. The entorhinal cortex is also barely stained. Therefore, we chose the CA1 region for our study. Secondly, in our previous study (Hackert et al.), we analyzed CA1 pyramidal neurons and aimed to compare the data from that study with the present results.

3) The meaning of the line# 196 is not clear. Maybe there is a word missing or there is an English error. 

Comment: We apologize for this prolonged and obviously misleading sentence. We have rephrased this paragraph.

Thank you very much

Mr. Frerker

Reviewer 2 Report

The study titled "Stereotactically injected human cerebrospinal fluid containing glutamic acid decarboxylase antibodies into the hippocampus of rats does not facilitate the development of spontaneous epileptic activity" by Frerker et al., is an interesting report and important to be published. The authors have shown using hippocampal slices and electrophysiological field potentials that contrary to the intuitive notion, GAD antibody does not facilitate the development of spontaneous activity. 

The manuscript is well written. However, there are some minor concerns: 

1) Please write the full version of RED in the abstract instead of just the abbreviation. 

2) It is not clear why the authors chose CA1 region for the recordings? Why not any other region of the Dentate? 

3) The meaning of the line# 196 is not clear. Maybe there is a word missing or there is an English error. 

Author Response

Dear Editor, and Dear Reviewer 2,

thank you very much for reading and evaluating our manuscript. Your comments were very useful to improve our manuscript. In the meantime, we had the opportunity to evaluate a GAD sample from another patient. Our new data suggests that this antibody could facilitate epileptic discharges in hippocampal slices, similar to the results of Ishida et al (1999). We included these novel data into our revised manuscript. Moreover, we repeated experiments showing the diffusion of an injected immunofluorescence marker dye.

We highlighted our changes in yellow. We hope this will help you to understand our changes.

Response to Reviewer Two

Comments and Suggestions for Authors

The study titled "Stereotactically injected human cerebrospinal fluid containing glutamic acid decarboxylase antibodies into the hippocampus of rats does not facilitate the development of spontaneous epileptic activity" by Frerker et al., is an interesting report and important to be published. The authors have shown using hippocampal slices and electrophysiological field potentials that contrary to the intuitive notion, GAD antibody does not facilitate the development of spontaneous activity. 

The manuscript is well written. However, there are some minor concerns: 

Comment: First, thank you very much for your evaluation of our manuscript. In the meantime, we had the opportunity to improve our manuscript. We received a CSF-sample from another patient. As a result, this anti-GAD facilitated the spontaneous epileptic activity in the hippocampus. This is an important finding, since anti-GAD does not necessarily mean that it has suppressive effects on the GAD enzyme. For example, there might be different epitopes on the GAD enzyme, and this might help explain why anti-GAD is associated with different diseases. We hope that our new results improve our manuscript. The changes we made are highlighted in yellow.

1) Please write the full version of RED in the abstract instead of just the abbreviation. 

Comment: We apologize that we forgot to write the full version of REDs. Now, we have introduced this abbreviation.

2) It is not clear why the authors chose CA1 region for the recordings? Why not any other region of the Dentate? 

Comment: Stanton et al (1986) showed that perfusion of hippocampal slices with Mg2+-free solution induced spontaneous epileptic activity in the dentate gyrus. To our knowledge, in their experiments, the connections to the entorhinal were intact. Mody et al (1987) then showed that REDs are induced in the CA1- and CA3-region, whereas the dentate gyrus did not. In these cases, the connection to the entorhinal cortex was severed. This implicates that intact entorhinal cortex is necessary for the induction of spontaneous epileptic activity. From previous studies and the current study (new figure 1 in the revised manuscript) we know that the antibody dispersion includes the upper blade of the dentate, but rarely the lower blade. The entorhinal cortex is also barely stained. Therefore, we chose the CA1 region for our study. Secondly, in our previous study (Hackert et al.), we analyzed CA1 pyramidal neurons and aimed to compare the data from that study with the present results.

3) The meaning of the line# 196 is not clear. Maybe there is a word missing or there is an English error. 

Comment: We apologize for this prolonged and obviously misleading sentence. We have rephrased this paragraph.